# Universal Agent for Disentangling Environments and Tasks

**Jiayuan Mao & Honghua Dong** *
The Institute for Theoretical Computer Science
Institute for Interdisciplinary Information Sciences
Tsinghua University
Beijing, China
{mjy14,dhh14}@mails.tsinghua.edu.cn

**Joseph J. Lim**
Department of Computer Science
University of Southern California
Los Angeles, USA
limjj@usc.edu

## Abstract

Recent state-of-the-art reinforcement learning algorithms are trained under the goal of excelling in one specific task. Hence, both environment and task specific knowledge are entangled into one framework. However, there are often scenarios where the environment (*e.g.* the physical world) is fixed while only the target task changes. Hence, borrowing the idea from hierarchical reinforcement learning, we propose a framework that disentangles task and environment specific knowledge by separating them into two units. The environment-specific unit handles how to move from one state to the target state; and the task-specific unit plans for the next target state given a specific task. The extensive results in simulators indicate that our method can efficiently separate and learn two independent units, and also adapt to a new task more efficiently than the state-of-the-art methods.

## 1 Introduction

Let's imagine ourselves learning how to play tennis for the first time. Even though we have never played tennis before, we already have a good understanding of agent and environment dynamics related to tennis. For example, we know how to move our arm from one position to another and that a ball will slow down and bounce back from the ground. Hence, we just need to learn the tennis specific knowledge (*e.g.* its game rule and a relationship between an arm control and a tennis racket). Just like this example, when we learn to complete a new task, we utilize the prior knowledge that is disentangled from the task and acquired over our lifetime.

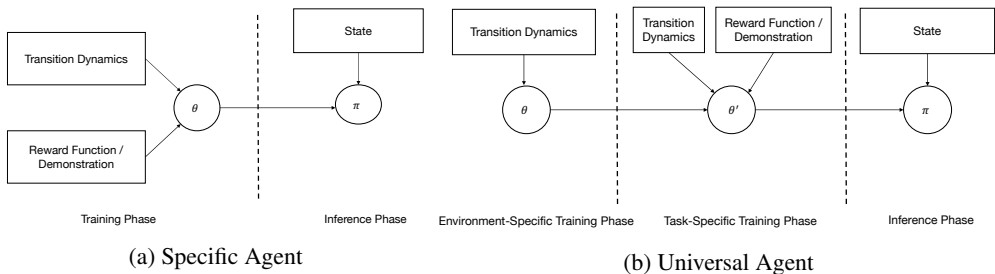

Figure 1: Our model disentangles environment-specific information (*e.g.* transition dynamics) and task-specific knowledge (*e.g.* task rewards) for training efficiency and interpretability.

From a reinforcement learning perspective, this brings a very interesting question – how can agents also obtain and utilize such disentangled prior knowledge about the environment? Most of today's deep reinforcement learning (DRL) models Mnih et al. (2015; 2016); Schulman et al. (2015a; 2017) are trained with entangled environment-specific knowledge (*e.g.* transition dynamics) and task-specific knowledge (*e.g.* rewards), as described in Figure 1a However, as described earlier, humans

---

*Work was done when Jiayuan and Honghua were visiting students at USC.

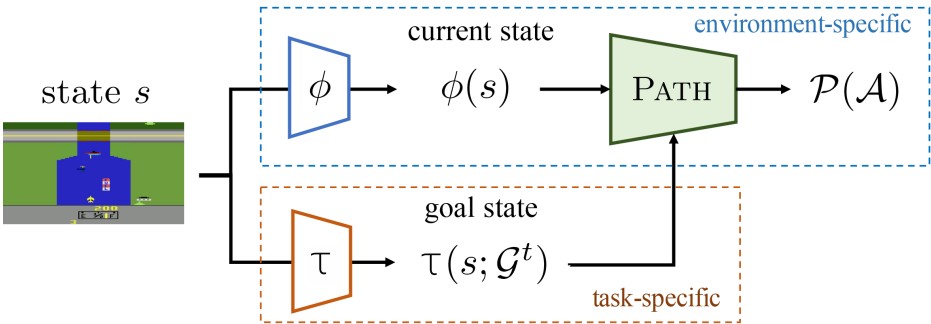

Figure 2: Proposed *universal agent*, which consists of three parts: a $\phi$ function mapping raw observation to feature space, a PATH function as an environment actor, and a $\tau$ function for future state planning.

have an innate ability to obtain a good understanding about the environment dynamics, and utilize them in a newly given task. Motivated from this, we introduce a new scheme to disentangle the learning procedure of task-independent transition dynamics and task-specific rewards, as described in Figure 2. This will help an agent to adapt to a new task more efficiently and also provides an extra interpretability.

The idea of disentangling a model into two components can be related to hierarchical RL approaches Sutton et al. (1999); Parr & Russell (1998); Oh et al. (2017). However, to the best of our knowledge, there has not been a work that separating units by the natural criteria of environment and task specific knowledge, for the goal of transfer learning.

To this end, we introduce a model that consists of two major units: a PATH function and a goal generator. This is illustrated in Figure 2. The key intuition is as the following. PATH function handles the environment specific knowledge, and a goal function handles the task specific knowledge. We design (1) PATH function to learn how to move from one state to another – a lower-level controller, which is independent from the task and only depends on the environment, and (2) the goal function $\tau$ to determine the next state given a target task – a higher-level planner. Thus, PATH function can be shared across different tasks as long as it is under the same environment (*e.g.* the physical world).

We evaluate our method to answer the following two questions: (1) how a good PATH unit can benefit the task learning, and (2) how efficient is our model for learning a new task in the same environment. We analyze the behavior of our method on various environments including a maze world and multiple Atari 2600 games. Our study shows that a good PATH unit can be trained, and our model has a faster convergence compared to the state-of-the-art method Mnih et al. (2016) in most of the tasks, especially on transfer learning tasks.

In summary, we introduce an RL model with disentangled units for task-specific and environment-specific knowledge. We demonstrate in multiple environments that our method learns environment-specific knowledge, which further enables an agent adapting to a new task in the same environment.

## 2 FRAMEWORK

Our goal in this paper is to introduce a reinforcement learning model that disentangles environment and task-specific knowledge. The advantage of such disentangling model is that we only need to learn task-specific knowledge when a goal of a task changes, or environment-specific knowledge when an environment of a task changes. This essentially is one form of generalization techniques across tasks or environments.

To this end, we introduce a modularized neural networks model – one unit handling a task-specific knowledge, and the other unit handling an environment-specific knowledge. For this, our model introduces two sub-functions: a goal function ($\tau$) and a path function (PATH). Our goal function ($\tau$) determines and generates the next target state for a given task (task-specific unit), and our path function (PATH) will provide the way to get to the target state (environment-specific unit). In this

setup, we only have to change $\tau$ function when the goal of a task changes (*e.g.* hitting 5 pins instead of 10 pins) in the same environment, because the way to get from one state to another state stays the same within the same environment (*i.e.* PATH function can remain the same). On the other hand, we only need to change PATH function if the environment changes (*e.g.* faster deceleration of a bowling ball) while the task is the same, because the way to go to one state from the current state changed (*e.g.* due to deceleration) while the goal of the task itself is the same.

## 2.1 PATH FUNCTION

We design our path function (PATH) to determine how to get to a state from another state, which represents environment-specific knowledge. Mathematically, we define: $\text{PATH}(s, s') \rightarrow \mathcal{P}(\mathcal{A})$. Given a *(starting state, target state)* pair, PATH function outputs a probability distribution over action space for the first action to take at state $s$ in order to reach state $s'$. For discrete action spaces $\mathcal{A} = \{a_1, a_2, \cdots, a_k\}$), the output is a categorical distribution, and for continuous action space: $\mathcal{A} = \mathbb{R}^k$, a multidimensional normal distribution with a spherical covariance is outputted (as Lillicrap et al. (2015)). To generate a trajectory from $s$ to $s'$, one can iteratively apply the PATH function and obtain a Markov chain.

Given the state space $\mathcal{S}$, PATH function is trained by sampling a starting state and a final state and teach the PATH function to generate the path from one to another using reward signal. Formally speaking, we define a series of game instances based on the original transition dynamics $s' = \text{Trans}(s, a)$[1], where $s$ is the state and $a$ is the action. For sampled state pair $(s, s') \in S' = \mathcal{S}^2$, we construct the following game instance $(S', \text{Trans}', \text{Reward}')$:

$$\text{Trans}'((s, s'), a) = (\text{Trans}(s, a), s'); \quad \text{Reward}'((s, s'), a) = \begin{cases} 1 & \text{if } \text{Trans}_e(s, a) = s' \\ -1 & \text{otherwise} \end{cases}.$$

We add the $-1$ term in reward function to enforce the actor to choose the shortest path getting from $s$ to $s'$. PATH function can be learned with any RL algorithms on the defined game instances.

The learning process of the PATH function is agnostic of any underlying task in the environment. When the state space is a prior knowledge or is easy to explore, the state pairs can be directly sampled.

For complex environments where the state space is not easily tractable, one can utilize exploration strategies (e.g., curiosity-driven RL Pathak et al. (2017)). Meanwhile, the PATH function, as a collection of "skills", can be jointly learned with one or more tasks. For example, PATH function and $\tau$ function are first jointly optimized on a source task. The PATH function is then reused in a new target task. In such cases, we sample state pairs from agents' historic experiences, which requires the restoration of an arbitrary state (see Atari games for detail). We leave other approaches for the PATH training as future work.

## 2.2 GOAL FUNCTION

We now design a goal function ($\tau$), which determines what the goal state should be for a given task, and thus task-specific knowledge. We define our goal function for a given task: $\tau(s; \theta^g) \approx \phi(s')$ to compose a target state in feature space given the current state. PATH function is then inferred to get the primitive action to be taken.

**Train with back-propagation** The simplest way to train the $\tau$ function is by back-propagation through the PATH module. As in A3C, actor corporates with a task-specific critic network which maintains a value estimation for any state $s$. Fixing the parameters for PATH $\theta^p$, the gradient for $\theta^g$ is computed as:

$$\nabla_{\theta^g} \log \pi(a|s; \theta^p, \theta^g) A(s, a; \theta^p, \theta^g, \theta^v) = \frac{\partial s'}{\partial \theta^g} \nabla_{s'} \log \text{PATH}(a|s, s'; \theta^p) A(s, a; \theta^p, \theta^g, \theta^v),$$

where $s' = \tau(s; \theta^g)$ and $A$ is the advantage computed by $\sum_{i=0}^{k-1} \gamma^i r_{t+i} + \gamma^k V(s_{t+k}; \theta^v) - V(s_t; \theta^v)$ and $\gamma$ is the discount factor Mnih et al. (2016).

---

[1]For simplicity here, environmental randomness is considered as a part of the transition dynamics. *E.g.* assuming $s$ contains the random seed of the environment.

**Reduce to a continuous control problem**    Another way to train the $\tau$ function is by viewing it as the response made by the agent and optimize it using the reward signal. Since the output lies on the manifold of $\phi(\mathcal{S})$, we can model it as a Gaussian distribution (as in continuous control RL). Any RL algorithms operated on continuous action space Schulman et al. (2015a;b); Lillicrap et al. (2015) can thus be applied to optimize the $\tau$ function.

A natural question would be: how to better choose the manifold of $\phi(\mathcal{S})$, which certainly affects the learning of $\tau$? For example, can we use $\beta$-VAE Higgins et al. (2016) which enforces the distribution of $\phi$'s output to be close a Gaussian distribution to help? In the experiment section, we will study several choices of the $\phi(\mathcal{S})$ space including auto-encoder Hinton & Salakhutdinov (2006), beta-VAE Higgins et al. (2016), forward dynamics Chiappa et al. (2017), inverse dynamics (one-step PATH function) Pathak et al. (2017), and the feature space jointly learned with PATH function.

Inspired by the recent advances in low-bits neural networks Rastegari et al. (2016); Zhou et al. (2016), to eliminate d.o.f. brought by the Gaussian distribution, we conduct experiments where we binarize the output coding of $\phi$. Specifically, we first train $\phi$ with a $\mathrm{tanh}$ activation at the last layer, then binarize the output, and finally fine-tune the $\phi$ again. By doing so, the output distribution of $\tau$ function becomes a binomial distribution.

**Notes on the $\tau$ function learning**    The major difference between two methods lies in the position where the sampling and the supervision of reward signals take place: at the output of PATH or the output of $\tau$. The first approach relies on PATH to update the planning made by $\tau$ using gradients back-propagated through the PATH module (environmental reward signals are used to optimize primitive actions that the model outputs). On the other hand, the second approach based on $\tau$ directly optimizes the state planning in the (latent) state space (environmental reward signals are directly used to optimize the goal state generated by $\tau$). We slightly prefer the first approach as it is easy to implement and is more stable when the (latent) state space is changing (because in some settings we are jointly training the PATH module and $\tau$ module).

## 2.3 TRAINING DETAILS

**Reinforcement learning model**    We implemented batched A3C as our base model following same settings as original paper Mnih et al. (2016). The discount factor is chosen as $\gamma = 0.99$ for all our experiments. We use a convolutional neural network (CNN) structure (2-4 layers depending on the task) instead of LSTM as feature extractor for $\phi$ and $\tau$, while PATH function is a 3-layer MLP. To preserve historical information, we concatenate four consecutive frames in channel dimension as input to our network (as state $s$). The critic network shares feature with $\tau$ (they are both task-specific) and has two extra fully-connected layers for value estimation.

**Curriculum learning for Path function**    The game defined in Section 2.1 is a game with very sparse reward. To accelerate the training process, we employ curriculum learning to help. In particular, the max distance between starting state and final state is initialized as one at the beginning of the training process. After every $K$ iterations of updating, the max distance is increased by one. This max distance can be either accurate (by finding the shortest path from $s$ to $s'$), heuristic (*e.g.*, Manhattan distance on a 2D maze), or determined by agents' experience (see experiments).

Compared to other pre-training methods or hierarchical RL methods, the proposed *universal agents* shows following two main advantages:

1. PATH function can be obtained without task specification. With only exploration (*e.g.*, naive random walk or curiosity-driven RL Pathak et al. (2017)), PATH function can be trained using the experience.
2. PATH function encodes no information about the task. When jointly learned on one or more tasks, PATH function shows better generalization to new tasks.

## 3 EXPERIMENTS

We begin our experiments with a small environment: Lava World. Rich ablation experiments are conduced to study the choice of feature space, optimization strategy and its knowledge transfer

ability. We further extends our result to a more realistic environment: Atari 2600 games. More experiments such as continuous control problem or imitation learning would be shown in the appendix.

## 3.1 LAVA WORLD

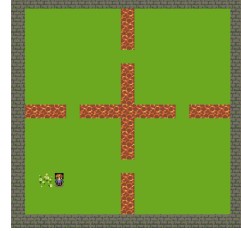

Lava world (Figure 3) is a famous 2D maze in reinforcement learning. The agent moves (in 4 directions) in the maze. The whole map is $15 \times 15$ and divided into 4 rooms. The grids with lava are obstacles to the agent. Therefore, to go from one room to another, the agent is forced to find the door connecting two rooms and go through it, which is harder than random generated 2D maze.

Figure 3: Lava World

**Path function**    The set of states in lava world is definite, we directly use randomly generated pairs of starting state and final state to train the path function. During this training phase, the agent understands how to move as well as how to go from one room to another.

We define the generalization ability of the PATH function as the ability to reach the target states whose distance is larger than training samples. We first test this by setting a max distance between the starting state and the final state during training but testing it on harder instances (the distance is longer). Figure 4a shows the result. When trained with very simple instances (distance is less than 7), the PATH function shows very limited generalization ability. When the training distance is increased to 7, it begins to generalize. This phenomenon was also witnessed in other experiments: too short training distance leads to pool generalization.

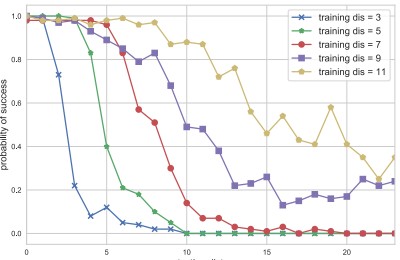

(a) Generalization analysis of PATH function.

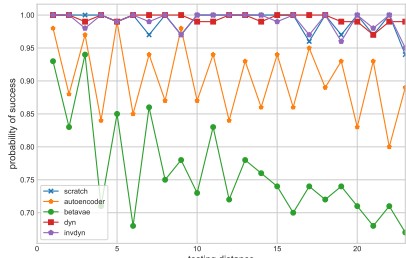

(b) The quality of path functions learned over different feature spaces.

Figure 4: Experiments showing the performance of the PATH function.

**Task: Reachability**    In this task, agents are asked to move to the target position presented on the map as a special mark. In all experiments, agents are provided with dense reward: when the agent is getting closer to the target position, it can receive positive reward. Shown in figure 5, *universal agent* demonstrates faster learning speed over pure A3C models with no prior knowledge. Furthermore, the learned $\tau$ function is well interpretable and we provide the visualization of the goal state composed by the agent in the appendix A.

In figure 5, the $\tau$ function is optimized by the gradients back-propagated through the PATH module, and the PATH function is almost "perfect". In other words, the PATH function can resolve the path between every pair of states. The goal generator only focuses on the goal state composition.

**Feature space and training strategy**    We now go deeper into the choice of feature space of $\phi$ and the training strategy for $\tau$.

We explored following feature spaces for the state encoder $\phi$:

- AutoEncoder: Encode the input state with an information bottleneck proposed in Hinton & Salakhutdinov (2006).
- $\beta$-VAE: Encode the input state with a disentangled coding by best compressing the information. We adjusted $\beta$ such that the information is reduced as much as possible under

the constraint that the reconstruction is perfect. From another point of view, $\beta$-VAE also enforces the feature space to be close to a Gaussian distribution. Higgins et al. (2016).

- Forward dynamics: Similar to Chiappa et al. (2017), we trained an neural network to approximate the dynamics of the environment: $p(s_{t+1}|s_t, a_t)$.

- Inverse dynamics: Also used by Pathak et al. (2017), inverse dynamics is an one-step PATH function: $p(a_t|s_t, s_{t+1})$.

- No pre-training (shown as "Scratch" in the figures). The feature space is jointly learned with PATH function.

We first train PATH function for certain $\phi$ space with sufficient amount of data. Shown in Figure 4b, auto-encoder and $\beta$-VAE perform worse than other feature spaces due to their ignorance of the underlying environment's dynamics. The interesting "odd-even" pattern, we hypothesis, is attributed to the feature space for pixel-level reconstruction.

For all feature space mentioned, we model the output distribution of $\tau$ function as multi-dimensional Gaussian distribution with a scalar variance $\sigma^2$. For their binarized variants, we model the distribution as binomial distribution. We also investigate two variants of the distribution modeling: Gaussian-Additive ($\phi(t) \approx \phi(s) + N(\mu, \sigma^2)$) and Binomial-Multiplicative ($\phi(t) = \phi(s) * \text{Binomial}(logits)$).

We use proximal policy optimization (PPO) Schulman et al. (2017) for the distributional $\tau$ learning. Shown in figure 6, Gaussian, and Gaussian-A have similar behavior while Binomial is typically a better choice than Binomial-B, probably because currently $\tau(s)$ does not depend on $\phi(s)$, which means the target state space composer has no information about the current state representation $\phi(s)$. Gaussian models perform well on all feature spaces with pixel-level

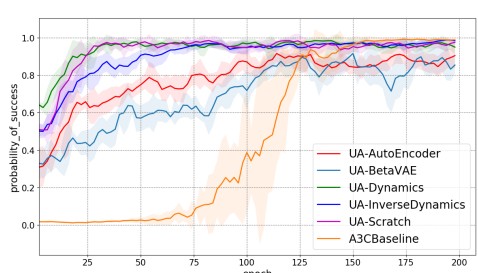

Figure 5: The rate of successfully reaching the final position in game Reachability. The universal agent model is trained through back-propagation.

information (*e.g.*, $\beta$-VAE and dynamics), while fails in the feature space with more high-level, discrete information (inverse dynamics). The hyper-parameters for the distributional models are generally more difficult to adjust, but might be useful in robotics settings where the input space itself is usually a continuous space (*e.g.*, joints of a robot). However, it shows no clear advantage over the results shown in figure 5 on the visual input games.

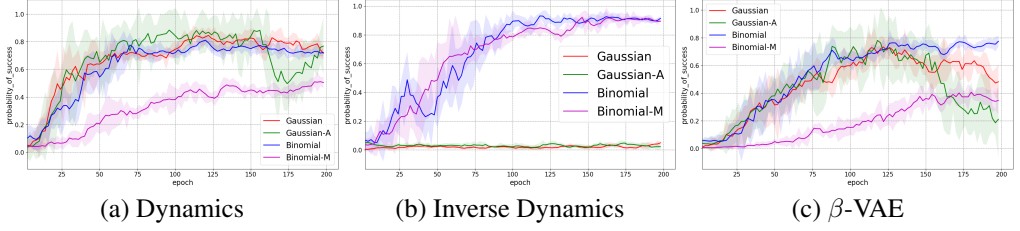

(a) Dynamics       (b) Inverse Dynamics       (c) $\beta$-VAE

Figure 6: Different feature spaces affect the learning of $\tau$ function

**Knowledge transfer via PATH function** We study two types of knowledge transfer: knowledge from exploration and knowledge from tasks. We implemented Pathak et al. (2017) as a no-reward RL explorer in unknown environments. The experience of the agents is collected for the PATH training. In no-reward exploration setting, PATH function is trained from scratch without pre-trained feature space. Since we can not exactly get the distance between two states in this setting (assuming we have no knowledge about the state space), when picking the starting state and the final state from the experience pool, we set the limit the number of steps that the explorer used as 15. With a same number of frames experienced ($\sim$ 32M frames), the PATH function can reach a comparable accuracy as the original curriculum learning correspondent. We compare the performance of two models

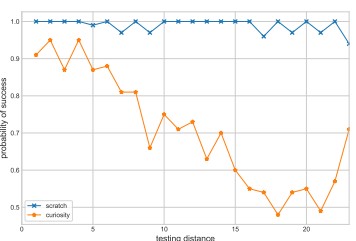

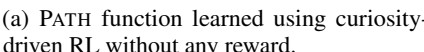

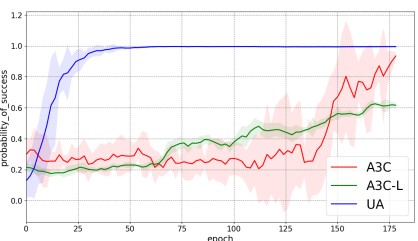

(a) PATH function learned using curiosity-driven RL without any reward.

(b) The probability of successfully finish the task during testing time in game Taxi. We only measure the probability of agents finishing the whole task.

Figure 7: Knowledge transfer via the PATH function.

in Figure 7a. When used with $\tau$ function for the reachability task, their performances are almost identical.

To study the transfer across multiple tasks, we define another task in the environment: Taxi. The agent now acts as a taxi driver, who first goes to one position to pick up the passenger (shown on the map), and then goes to another place. The target position will be provided to the agent by two one-hot vectors representing the $(x, y)$ coordinates after our agent picks up the passenger. In order to complete the task, the agent learns how to interpret the task representation (one-hot vectors).

We compare the performance of *universal agent* with both A3C baseline trained from scratch as well as A3C model pre-trained for reachability game (A3C-L) in Figure 7b. Even though the first phase of the task is exactly the same as reachability game, *universal agent* excels in the task interpretation and outperforms both baselines.

## 3.2 ATARI 2600 GAMES

We tested our framework on a subset of Atari 2600 games, a challenging RL testbed that presents agents with a high-dimensional visual input ($210 \times 160$ RGB frames) and a diverse set of tasks which are even difficult for human players.

**Feature spaces and PATH function** A critical feature of Atari games is the difficulty of exploration, which is attributed to the complexity of the environment as well as the intricate operation. We use state restoration for the PATH learning, starting state and targeting state are sampled from agents' experience performing tasks or exploration. The state restoration here is an extra requirement for the learning, and we believe that techniques for PATH training without state restoration can be separately addressed.

Also, as the state space is not known as a prior knowledge for Atari games, we directly optimize the state encoder $\phi$ jointly with the PATH function.

We prefer to train a PATH function jointly with tasks. We first chose 8 typical Atari games to validate how PATH function can be learned jointly and show its effectiveness in the appendix D and visualization in the appedix F. Also, we provide analysis on using pure exploration for PATH function in the appendix E.

**Knowledge transfer from other tasks** *Universal agent* is capable of efficient transfer learning. We further define some new tasks for four of the Atari games. The task specifications are included in the appendix G. All new tasks have large differences with the corresponding original task in the game (*E.g.*, in original Riverraid, the agents should fire the boats. While in the new task, the agents should avoid firing any boats.) In Figure 8, we demonstrate how our *universal agent* can benefit from the pre-trained PATH function in the original task (jointly learned with a $\tau$ function for the original task). For a fair comparison, in A3C-L, we load all the weights (except the last fully-connected layers for policy and value) from the model trained on original task, while A3C represents A3C trained from scratch.

In both Alien-new and Riverraid-new, UA with pre-trained PATH function outperforms A3C baselines. We also show two more games where UA fails to perform very well. In game UpNDown-new, A3C

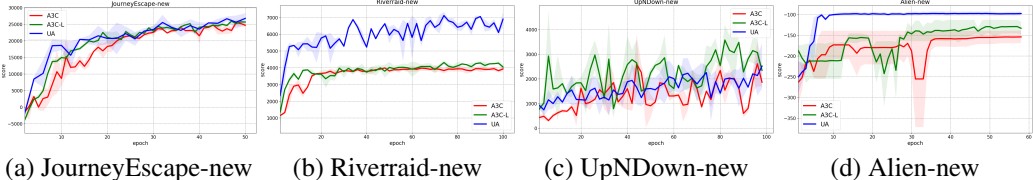

| (a) JourneyEscape-new | (b) Riverraid-new | (c) UpNDown-new | (d) Alien-new |

Figure 8: Comparison between A3C, A3C with pre-trained model on the original task, and UA with pre-trained PATH function. The translucent regions indicate the standard deviations of multiple runs. All environments are based on the latest versions of Gym Atari.

succeeds in finding a bug in the game, and use it to escape from other cars. While performing slightly worse than A3C, UA chooses to jump over other cars to avoid being hit, which is more reasonable and rational. In JourneyEscape-new, both UA and A3C-L perform better than A3C trained from scratch.

**Task transfer from multiple sources**  We demonstrate the ability of knowledge transfer from multiple tasks in two Atari games. A set of pre-trained A3C agents for different tasks in same environments are employed to collect experiences for our agent in the PATH function training phase. This method essentially "distills PATHfrom multiple agents". The task specifications can be found in the appendix. The learning procedure corresponds to humans' life-long learning from multiple tasks in the same environment. With this prior knowledge, agents are trained on completely new tasks with PATH function given and fixed. Results for A3C-S and A3C-L are also provided as baseline models.

Another implicit advantage of *universal agent* is that PATH functions, as sets of skills, will filter out meaningless actions. When we extract PATH function from pre-trained models, PATH function only learns how to reach those meaningful states. This helps the exploration and exploitation for the agents in new target tasks.

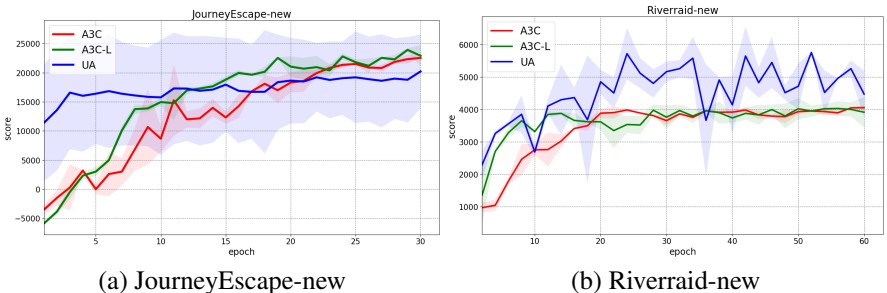

| (a) JourneyEscape-new | (b) Riverraid-new |

Figure 9: Comparison between A3C, A3C with pre-trained model on one of source tasks, and UA with pre-trained PATH function from multiple source tasks. The translucent regions indicate the standard deviations of multiple runs. All environments are based on the latest versions of Gym Atari.

In these two environments, *universal agent* shows faster convergence and better performance than a3c with pre-trained models on another task. Currently, the results are not completely comparable with the method for transfer from a single source (*e.g.*, for the Riverraid task). How to effectively learn the PATH function from several tasks, either in a parallel or a sequential manner, falls into the category of continual learning and life-long learning Ring (1994; 1997). We leave this as a future work.

## 4  RELATED WORKS AND DISCUSSION

**Model-based reinforcement learning**  Typical model-based RL frameworks aim to model the dynamics of the environment. They usually involve search-based algorithms (*e.g.*, Monte Carlo Tree Search) as part of the policy Sutton & Barto (1998) (*e.g.*, Alpha GO with a simulator Silver et al. (2017), Scheme-networks with learned forward dynamics Kansky et al. (2017)). In this paper, we address the problem of representing the knowledge about the environment by learning skills (how to reach a given state). As a cognitive science motivation described in Graziano (2006), humans also store the knowledge of movements or actions by their end-states. Most importantly, this knowledge can be easily utilized by a task-specific module (the goal generator) to exploit novel tasks.

**Multi-task learning and transfer learning**  Multi-task learning and transfer learning Devin et al. (2016); Andreas et al. (2016); Taylor & Stone (2009) provide approaches to transfer knowledge among multiple agents. The methods include the decomposition of value function or task and direct multi-task learning where agents learn several tasks jointly. Contrary to them, *universal agent* is able to obtain the environment specific knowledge without any specific task supervision.

**Hierarchical reinforcement learning**  In standard hierarchical RL (HRL) Sutton et al. (1999); Hernandez-Gardiol & Mahadevan (2001); Kulkarni et al. (2016); Fox et al. (2017) a set of subtasks are defined by a hyper-controller. Low-level controllers receive subtask signals from the hyper-controller and output primitive actions. Both low-level controllers and the hyper-controller (sometimes called meta-controller) learn from a specific task. Contrary to them, the proposed PATH function can be trained without supervision and provides a more general interface for potential tasks. Unlike Schaul et al. (2015); Vezhnevets et al. (2017), the $\tau$ function does not perform option selection but directly composes target state since a general-purpose PATH function is utilized. We do not follow typical HRL methods where once the subtask is selected, the low-level controller will be executed for multiple time steps. In *universal agent*, for simplicity now, $\tau$ function plans the future state at every time step. We leave its adaption to HRL frameworks as a future work.

**Instruction-based reinforcement learning**  In Oh et al. (2017), instruction are provided to agents as task description. Trained with the ability to interpret instructions, agents can perform zero-shot generalization on new tasks. However, the performance of this method is still restricted by the set of tasks the agents are trained with. *Universal agent* addresses the generalization among multiple tasks by a general PATH function.

**Path function**  PATH function is an instance of *universal value function approximator* Schaul et al. (2015). While a set of works use the prediction of future state as knowledge representation for RL agents Chiappa et al. (2017); Dosovitskiy & Koltun (2016) (*i.e.*, trying to approximate the transition dynamics $p(s_{t+1}|s_t, a_t)$), the proposed PATH function can be viewed as an inverse and a high-level abstract version of the approximated transition dynamics. From another point of view, PATH function is a variant of the feasibility function which measures the reachability from one state to another. An important property of PATH function is the ignorance of any specific tasks. Therefore, PATH function can be shared among different agents working on different tasks but in the same environment.

**Imitation learning**  Imitation learning Hester et al. (2017); Duan et al. (2017) considers the problem of deriving a policy from examples provided by a teacher. Contrary to Reinforcement Learning (RL) where policy derived from experience, it provides an intuitive way to communicate task information as human. A common solution is to performs supervised learning from observations to actions Pomerleau (1989); Ross et al. (2011). However, purely supervised learning suffers from the lack of knowledge about domain dynamics, which leads to bad generalization when the environment is complex Hester et al. (2017). We show in the appendix how we incorporate prior knowledge of environment into an imitation learning framework and improve its generalization and robustness.

## 5  CONCLUSION AND FUTURE WORKS

We present a new reinforcement learning scheme that disentangles the learning procedure of task-independent transition dynamics and task-specific rewards. The main advantage of this is efficiency of task adaptation and interpretability. For this we simply introduce two major units: a PATH function and a goal generator. Our study shows that a good PATH unit can be trained, and our model outperforms the state-of-the-art method Mnih et al. (2016) in most of tasks, especially on transfer learning tasks.

The proposed framework is a novel step towards the knowledge representation learning for deep reinforcement learning (DRL). There are a variety of future research directions. For example, how PATH function can be learned (*e.g.*, in a continual learning manner Ring (1994), with less requirement such as state restoration), how it can better cooperate with the goal generator (*e.g.*, incorporating explicit future planning) and how it can be used for other tasks (*e.g.*, learning from demonstration).

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

# Supplementary Materials for Universal Agent

## A    INTERPRETABLE GOAL STATE IN THE REACHABLITY GAME

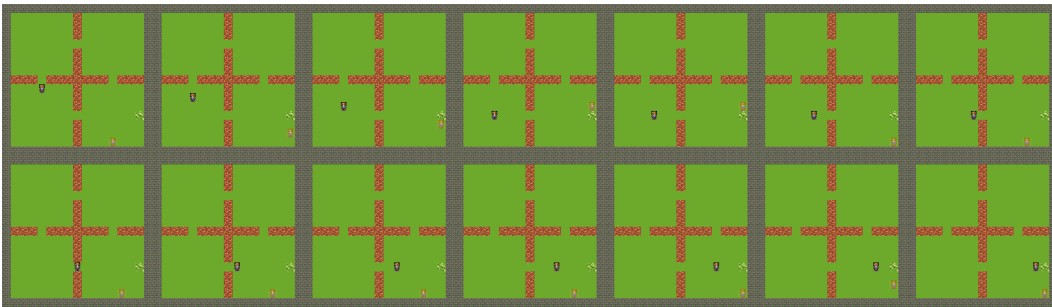

Figure 10: Visualization of the goal state composed by universal agent

Given the feature space of all states, we can visualize the goal state generated by $\tau$ function by link it to the nearest real state in feature space. The shadow agent in Figure 10 shows the corresponding goal position when the agent is at its real position. The *universal agent* model can thus dramatically increase the interpretability of the agent's play in certain games.

## B  IMITATION LEARNING IN LAVAWORLD

In imitation learning (IL), task supervision is provided by the demonstration coming from either human experts or other pre-trained agents. A straightforward way is to minimize $\mathbb{E}_{(s,a)\in\mathcal{D}}\log\pi(a|s;\theta)$. Given a set of demonstration plays (sequences of state-action pair $\mathcal{D}$), we can add extra supervision signal on *Universal Agent* to make it better understand the goal.

$$\mathcal{L}' = \mathcal{L} + \alpha\,\mathbb{E}_{\text{demo}}\text{KL}[\tau(s_t)|s_{t+k}],$$

where $s_t, s_{t+k}$ are two sampled state from a trajectory in $\mathcal{D}$. $k$ is a planning horizon factor.

While the primitive action policy was obtained in environment-specific training phase, agents are forced to understand how to plan future state (what should happen) instead of primitive actions (what should be done). We show empirically that *universal agent* demonstrate better generalization and robustness over pure imitation learning.

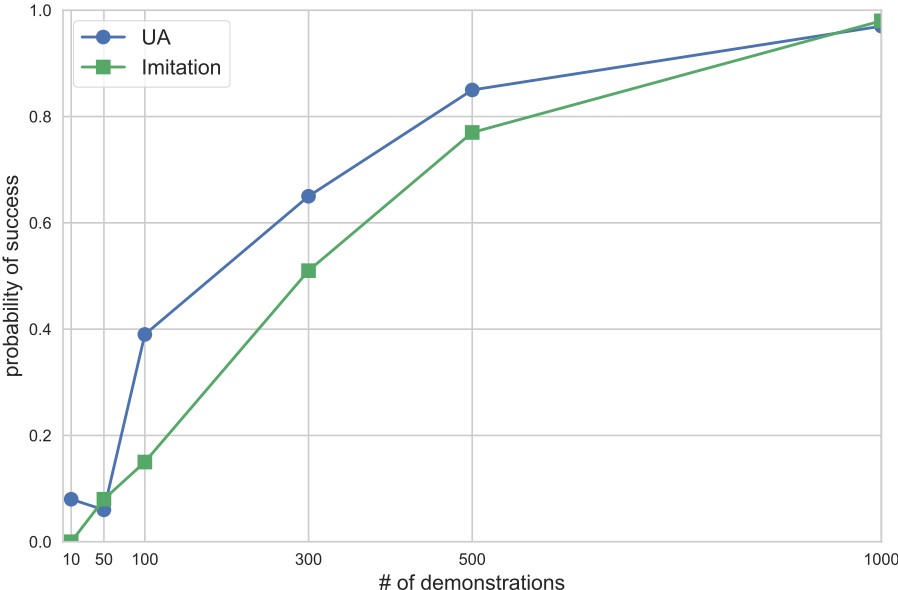

Figure 11: The probability of successfully reaching the final position during testing time in game Reachability by pure imitation learning.

Figure 11 shows the empirical analysis of agents' performance in the game with respect to the number of demonstration plays. With a limited number of demonstration plays, *universal agent* interprets the task and shows better generalization than agents trained from scratch. Figure **??** illustrates consistent results for imitation learning + RL.

## C   IMINATAION LEARNING IN CONTINUOUS CONTROL

We choose Lunar Lander (Figure 12 as environment to evaluate *universal agent* in continuous control setting Lillicrap et al. (2015); Schulman et al. (2015a;b). In this environment, the agent controls a spacecraft with three engines (main, left and right) to land safely on the moon. Different from discrete control version where the agent can only choose the fire each engine or not, in continuous control, an engine can be fired partly, which enables more meticulous control. The reward is determined by whether the agent land successfully as well as the fuel consumed. The randomness of the environment comes from the configuration of the land as well as initial position and velocity of the lander. The baseline model is the continuous control version of A3C as in Mnih et al. (2016).

In lunar lander, the state space is rather difficult to explore by random actions. Thus we make use of a pre-trained A3C model for the landing task and other heuristic agents for imitation learning (described later) to provide demonstration plays for our agent in the PATH function training phase. To ensure the generality of learned PATH function, a large noise is added to the action performed by the demonstrators ($N(0, 0.6)$ in our setting, compared to the valid value interval $[-1, 1]$).

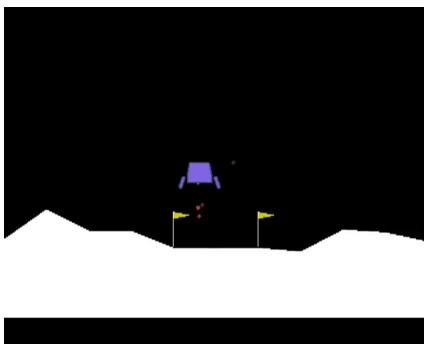

Figure 12: Lunar Lander

The Lunar Lander is a typical task where reward function is hard to design, especially for special tasks. To show the generalization ability, we design three new tasks for imitation learning: hover in the center, fly away from left-middle, and swing near the center. We provide the agent with a small set of demonstration plays (1 for all tasks) generated by heuristic players. The imitation learning baseline is a CNN model trained by minimizing the KL-divergence between policy $\pi(a'|s)$ with one-hot policy $a$ for all demonstration pairs $(s, a)$. Shown in Figure 13, *universal agent* with pre-trained PATH function outperforms pure imitation learning model in generalization and robustness.

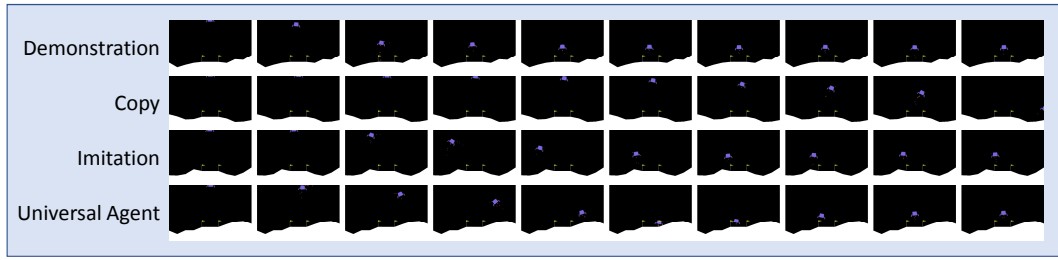

(a) Lunar Lander extra task: Hover

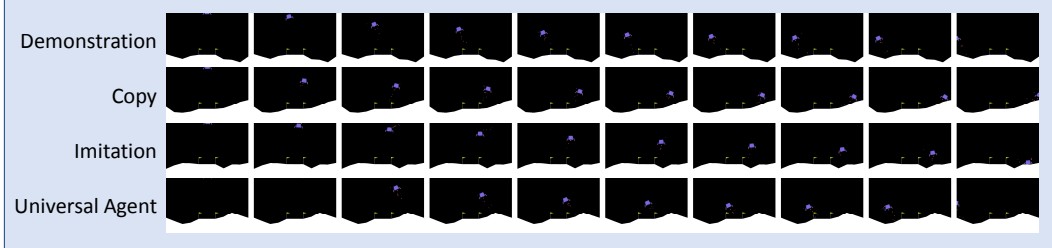

(b) Lunar Lander extra task: Fly-away from left-middle

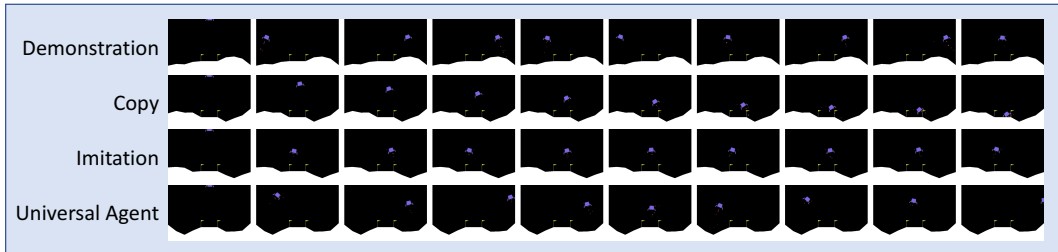

(c) Lunar Lander extra task: Swing

Figure 13: Learn from only demonstration in Lunar Lander. "Copy" represents agents which directly copy demonstration's actions as baseline.

# D  JOINT TRAINING OF PATH FUNCTION AND $\tau$ FUNCTION IN ATARI GAMES

We modify the training pipeline for A3C and include two phases: PATH training and $\tau$ training, which are performed alternately. At first, with no knowledge about the state space, agents gather experience by random actions, which is further used for the training of PATH function. Then, with PATH function fixed, $\tau$ is trained for the task. After a fixed number of iterations, since the agent has explored more in the state space, PATH function is trained again based on all experiences collected during the exploitation. And the max number of steps for PATH function is chosen as 15 for all experiments.

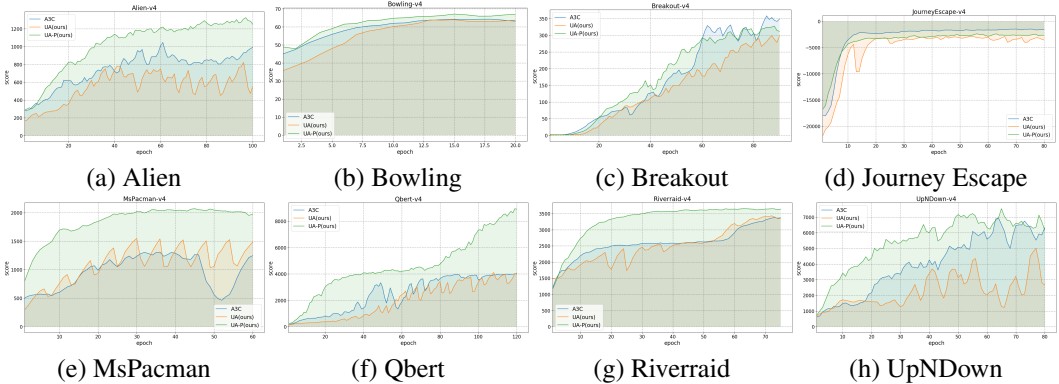

| (a) Alien | (b) Bowling | (c) Breakout | (d) Journey Escape |
|---|---|---|---|
| (e) MsPacman | (f) Qbert | (g) Riverraid | (h) UpNDown |

Figure 14: Comparison between A3C, jointly trained UA, and UA with pre-trained PATH function. With a pre-trained PATH function, a faster convergence speed in universal agent is witnessed.

# E  KNOWLEDGE TRANSFER FROM EXPLORATION IN ATARI GAMES

We study how *universal agent* can benefit from a PATH function trained by the experience of pure exploration without any reward. Similar to Lava World experiments, we employ the curiosity-driven model to collect experience.

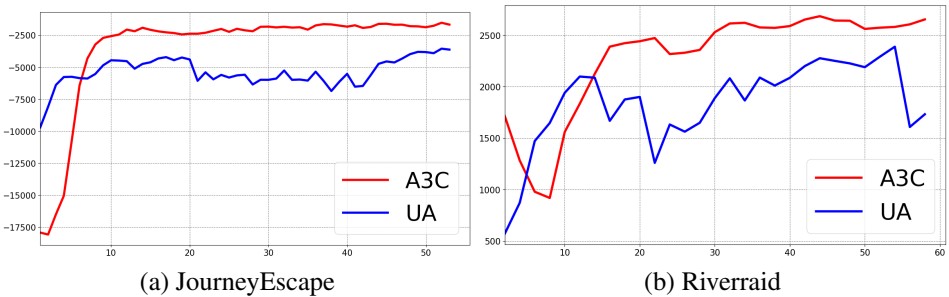

(a) JourneyEscape          (b) Riverraid

Figure 15: Knowledge transfer from exploration in Atari games

In Figure 15, with PATH function learned during exploration, the agents showed faster learning speed on the given task initially. However, their performance (score) is typically worse than the A3C model. In both tasks, we find that the agents are able to control themselves, but fail to effectively avoid colliding with objects which will lower the score. In path training, because of the absence of reward signal, the PATH function fails to learn how to avoid certain "bad" objects.

## F VISUALIZATION FOR THE LEARNED PATH FUNCTION

We visualize the Jacobian w.r.t. both inputs to the PATH function as the saliency map. This shows how PATH function makes the decision on the next action to be taken. We provide this analysis aiming at showing the generality and interpretability of the learned PATH function. *I.e.*, it does learn the dynamics of the environment and can be utilized by the goal generator later in novel tasks.

In both games, the PATH function learns how to move the agent and thus naturally take its current and target position into consideration. Moreover, neurons activated on some objects in the scene, especially for the game Riverraid. In game Riverraid, as the background is moving continuously, agents do not have any indicator for their positions. We conjecture that the PATH function utilize the position of other objects to locate the agent.

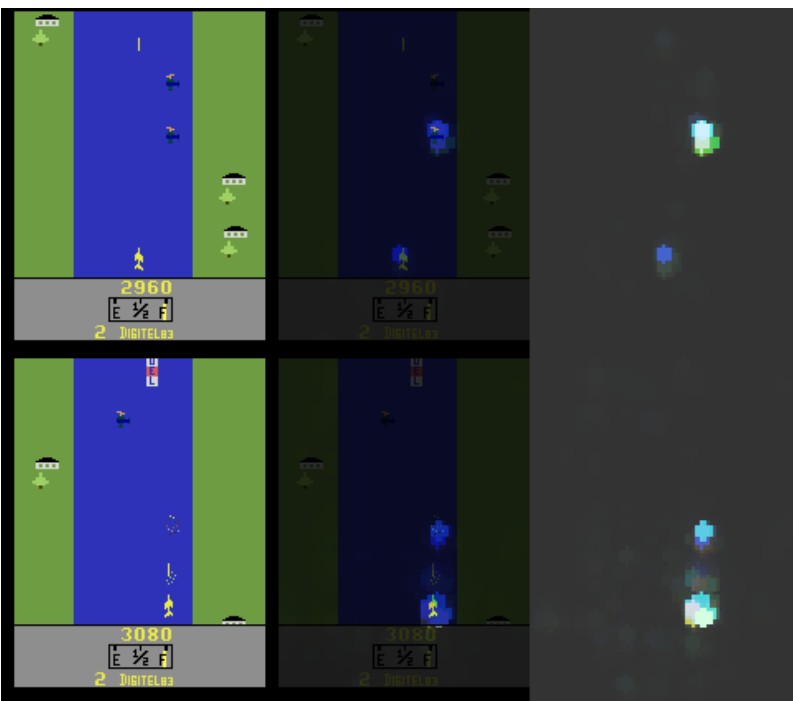

Figure 16: Visualizations of the saliency maps for the PATH function w.r.t. both inputs in the game Riverraid Scene #1. Columns (left-to-right): input state, the saliency map, the Jacobian. First row: current state. Second row: next state (with certain distances).

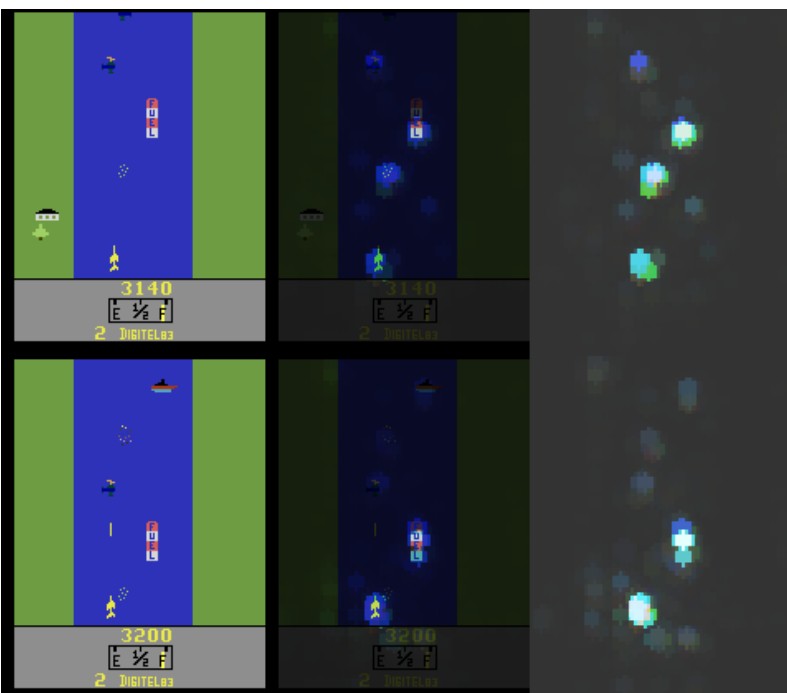

Figure 17: Visualizations of the saliency maps for the PATH function w.r.t. both inputs in the game Riverraid Scene #2. Columns (left-to-right): input state, the saliency map, the Jacobian. First row: current state. Second row: next state (with certain distances).

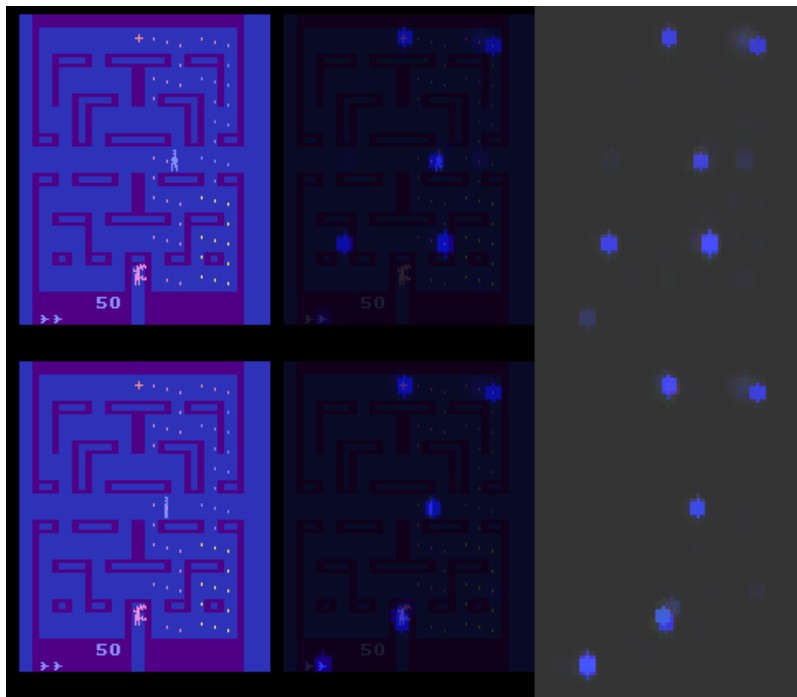

Figure 18: Visualizations of the saliency maps for the PATH function w.r.t. both inputs in the game Alien Scene #1. Columns (left-to-right): input state, the saliency map, the Jacobian. First row: current state. Second row: next state (with certain distances).

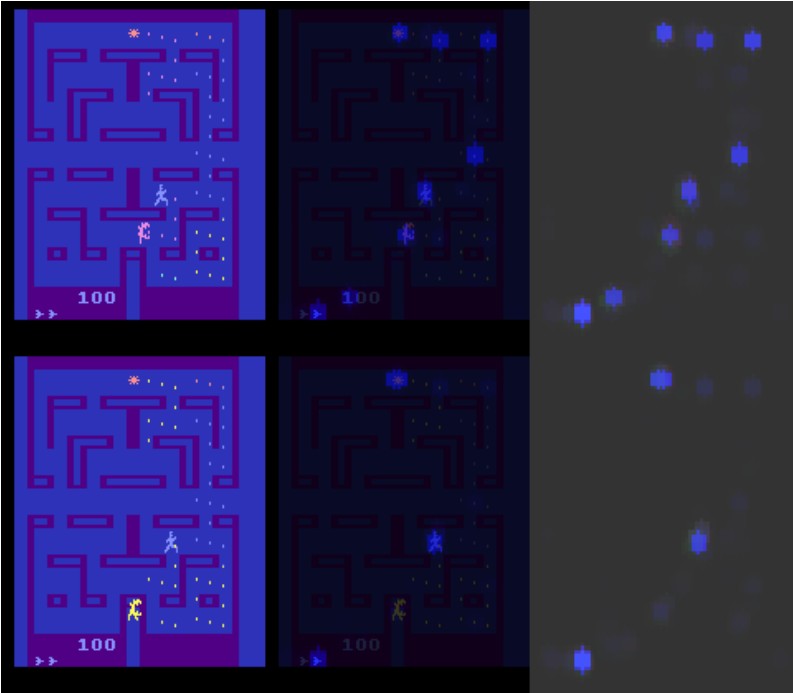

Figure 19: Visualizations of the saliency maps for the PATH function w.r.t. both inputs in the game Alien Scene #2. Columns (left-to-right): input state, the saliency map, the Jacobian. First row: current state. Second row: next state (with certain distances).

## G   SPEC OF NEW TASKS DEFINED OVER ATARI GAMES

**New tasks for task transfer from single source**

[1] Riverraid-new: Shooting boat will now get penalty, instead of reward (*i.e.* get reward -30 instead of 30).

[2] UpNDown-new: Live longer, but try not to squash other cars (in original game, squashing cars gets reward 400, now it gets rewards -400).

[3] Alien-new: Try to find the guard and dies as soon as possible, receive -1 reward each step (in original game, you need to avoid meeting any guards and collect score).

[4] JourneyEscape-new: Touching red-heart shaped objects will get reward 300, instead of getting penalty (-300).

**tasks for task transfer from multiple sources**

[1.1] Riverraid-bad-boat: Shooting boat will now get penalty, instead of reward (*i.e.* get reward -30 instead of 30).

[1.2] Riverraid-bad-helicopter: Shooting helicopter will now get penalty, instead of reward (*i.e.* get reward -60 instead of 60)

[1.3] Riverraid-bad-fuel: Shooting fuels will now get penalty, instead of reward (*i.e.* get reward -80 instead of 80).

[1.4] Riverraid-original: Original task.

[2.1] JourneyEscape-good-heart: Touching red-heart shaped objects will get reward 300, instead of penalty (-300).

[2.2] JourneyEscape-good-ball: Touching yellow-ball shaped objects will get reward 600, instead of penalty (-600).

[2.3] JourneyEscape-good-head: Touching head shaped objects will get reward 2000, instead of penalty (-2000).

[2.4] JourneyEscape-original: Original task.

We choose [1.1]Riverraid-bad-boat and [2.1]JourneyEscape-good-heart as our target task, and PATH function is pre-trained with demonstrations from other three tasks. The $\tau$ function is fine-tuned in one of the source tasks. Then we transfer the $\tau$ function to target task and get the result.

