# OpenReview forum: "Universal Agent for Disentangling Environments and Tasks"
_ICLR.cc/2018/Conference — Accept (Poster)_

### Official Review · AnonReviewer2 · 2017-11-27
**Novel continuous hierarchical like RL proposal demonstrated on concrete problems  but needs more details and focus on less material**

**Rating:** 6
**Confidence:** 3

**Review:**

The authors propose to decompose reinforcement learning into a PATH function that can learn how to solve reusable sub-goals an agent might have in a specific environment and a GOAL function that chooses subgoals in order to solve a specific task in the environment using path segments. So I guess it can be thought of as a kind of hierarchical RL.
The exposition of the model architecture could use some additional detail to clarify some steps and possibly fix some minor errors (see below). I would prefer less material but better explained. I had to read a lot of sections more than once and use details across sections to fill in gaps. The paper could be more focused around a single scientific question: does the PATH function as formulated help?

The authors do provide a novel formulation and demonstrate the gains on a variety of concrete problems taken form the literature. I also like that they try to design experiments to understand the role of specific parts of the proposed architecture.

The graphs are WAY TOO SMALL to read. Figure #s are missing off several figures.


MODEL & ARCHITECTURE

The PATH function given a current state s and a goal state s', returns a distribution over the best first action to take to get to the goal P(A).  ( If the goal state s’ was just the next state, then this would just be a dynamics model and this would be model-based learning? So I assume there are multiple steps between s and s’?).

At the beginning of section 2.1, I think the authors suggest the PATH function could be pre-trained independently by sampling a random state in the state space to be the initial state and a second random state to be the goal state and then using an RL algorithm to find a path.

Presumably, once one had found a path ( (s, a0), (s1, a1), (s2, a2), …, (sn-1,an-1),  s’ ) one could then train the PATH policy on the triple (s, s’, a0) ? This seems like a pretty intense process: solving some representative subset of all possible RL problems for a particular environment … Maybe one choses s and s’ so they are not too far away from each other (the experimental section later confirms this distance is >= 7. Maybe bring this detail forward)?

The expression Trans’( (s,s’), a) = (Trans(s,a), s’) was confusing.  I think the idea here is that the expression
Trans’(  (s,s’) , a ) represents the n-step transition function and ‘a' represents the first action?

The second step is to train the goal function for a specific task. So I gather our policy takes the form of a composed function and the chain rule gives close to their expression in 2.2

    PATH(  s,  Tau( s, th^g ),  a ; th^p )

    d / { d th^g }  PATH(  s,  Tau( s, th^g ),  a ; th^p )

         =  {d / d {s’ }  PATH } ( s,  Tau( s, th^g ),  a )    d / {d th^g}  Tau( s, th^g)

What is confusing is that they define

    A( s, a, th^p, th^g, th^v ) = sum_i   gamma^i  r_{t+1}  +  gamma^k  V(  s_{t+k}  ;  th^v  )   -   V( s_t ;  th^v )

The left side contains th^p and th^g, but the right side does not. Should these parameters be take out of the n-step advantage function A?

The second alternative for training the goal function tau seems confusing. I get that tau is going to be constrained by whatever representation PATH function was trained on and that this representation might affect the overall performance - performance.  I didn’t get the contrast with method one.  How do we treat the output of Tau as an action? Are you thinking of the gradient coming back through PATH as a reward signal? More detail here would be helpful.


EXPERIMENTS:

Lavaworld: authors show that pretraining the PATH function on longer 7-11 step policies leads to better performance
when given a specific Lava world problem to solve.  So the PATH function helps and longer paths are better. This seems reasonable. What is the upper bound on the size of PATH lengths you can train?

Reachability: authors show that different ways of abstracting the state s into a vector encoding affect the performance of the system.  From a scientific point of view, this seems orthogonal to the point of the paper, though is relevant if you were trying to build a system.

Taxi: the authors train the PATH problem on reachability and then show that it works for TAXI. This isn’t too surprising. Both picking up the passenger (reachability) and dropping them off somewhere are essentially the same task: moving to a point. It is interesting that the Task function is able to encode the higher level structure of the TAXI problem’s two phases.

Another task you could try is to learn to perform the same task in two different environments. Perhaps the TAXI problem, but you have two different taxis that require different actions in order to execute the same path in state space. This would require a phi(s) function that is trained in a way that doesn’t depend on the action a.

ATARI 2600 games: I am not sure what state restoration is.  Is this where you artificially return an agent to a state that would normally be hard to reach? The authors show that UA results in gains on several of the games.

The authors also demonstrate that using multiple agents with different policies can be used to collect training examples for the PATH function that improve its utility over training examples collected by a single agent policy.

RELATED WORK:

Good contrast to hierarchical learning: we don’t have switching regimes here between high-level options

I don’t understand why the authors say the PATH function can be viewed as an inverse? Oh - now I get it.
Because it takes an extended n-step transition and generates an action.

---

### Official Review · AnonReviewer1 · 2017-11-27
**Interesting alternative architecture on par with standard A3C policy representations.**

**Rating:** 7
**Confidence:** 4

**Review:**

In this paper a modular architecture is proposed with the aim of separating environment specific (dynamics) knowledge and task-specific knowledge into different modules. Several complex but discrete control tasks, with relatively small action spaces, are cast as continuous control problems, and the task specific module is trained to produce non-linear representations of goals in the domain of transformed high-dimensional inputs.

Pros
- “Monolithic” policy representations can make it difficult to reuse or jointly represent policies for related tasks in the same environment; a modular architecture is hence desirable.
- An extensive study of methods for dimensionality reduction is performed for a task with sparse rewards.
- Despite all the suggestions and questions below, the method is clearly on par with standard A3C across a wide range of tasks, which makes it an attractive architecture to explore further.

Cons
- In general, learning a Path function could very well turn out to be no simpler than learning a good policy for the task at hand. I have 2 main concerns:
The data required for learning a good Path function may include similar states to those visited by some optimal policy. However, there is no such guarantee for random walks; indeed, for most Atari games which have several levels, random policies don’t reach beyond the first level, so I don’t see how a Path function would be informative beyond the ‘portions’ of the state space which were visited by policies used to collect data.
Hence, several policies which are better than random are likely to be required for sampling this data, in general. In my mind this creates a chicken-and-egg issue: how to get the data, to learn the right Path function which does not make it impossible to still reach optimal performance on the task at hand? How can we ensure that some optimal policy can still be represented using appropriate Goal function outputs? I don’t see this as a given in the current formulation.
- Although the method is atypical compared to standard HRL approaches, the same pitfalls may apply, especially that of ‘option collapse’: given a fixed Path function, the Goal function need only figure out which goal state outputs almost always lead to the same output action in the original action space, irrespective of the current state input phi(s), and hence bypass the Path function altogether; then, the role of phi(s) could be taken by tau(s), and we would end up with the original RL problem but in an arguably noisier (and continuous) action space. I recommend comparing the Jacobian w.r.t the phi(s) and tau(s) inputs to the Path function using saliency maps [1, 2]; alternatively, evaluating final policies with out of date input states s to phi, and the correct tau(s) inputs to Path function should degrade performance severely if it playing the role assumed. Same goes for using a running average of phi(s) and the correct tau(s) in final policies.
- The ability to use state restoration for Path function learning is actually introducing a strong extra assumption compared to standard A3C, which does not technically require it. For cheap emulators and fully deterministic games (Atari) this assumption holds, but in general restoring expensive, stochastic environments to some state is hard (e.g. robot arms playing ping-pong, ball at given x, y, z above the table, with given velocity vector).
- If reported results are single runs, please replace with averages over several runs, e.g. a few random seeds. Given the variance in deep RL training curves, it is hard to make definitive claims from single runs. If curves are already averages over several experiment repeats, some form of error bars or variance plot would also be informative.
- How much data was actually used to learn the Path function in each case? If the amount is significant compared to task-specific training, then UA/A3C-L curves should start later than standard A3C curves, by that amount of data.


References
[1] Simonyan, K., Vedaldi, A., and Zisserman, A. Deep inside
convolutional networks: Visualising image classification
models and saliency maps. arXiv preprint arXiv:1312.6034, 2013.
[2] Z Wang, T Schaul, M Hessel, H Van Hasselt, M Lanctot, N De Freitas, Dueling network architectures for deep reinforcement learning arXiv preprint arXiv:1511.06581

---

### Official Review · AnonReviewer3 · 2017-11-29
**Interesting paper**

**Rating:** 6
**Confidence:** 3

**Review:**

Thank you for the submission. It was an interesting read. Here are a few comments:

I think when talking about modelling the dynamics of the world, it is natural to discuss world models and model based RL, which also tries to explicitly take advantage of the separation between the dynamics of the world and the reward scheme. Granted, most world model also try to predict the reward. I’m not sure there is something specific I’m proposing here, I do understand the value of the formulation given in the work, I just find it strange that model based RL is not mention at all in the paper.

I think reading the paper, it should be much clearer how the embedding is computed for Atari, and how this choice was made. Going through the paper I’m not sure I know how this latent space is constructed. This however should be quite important. The goal function tries to predict states in this latent space. So the simpler the structure of this latent space, the easier it should be to train a goal function, and hence quickly adapt to the current reward scheme.

In complex environments learning the PATH network is far from easy. I.e. random walks will not expose the model to most states of the environment (and dynamics). Curiosity-driven RL can be quite inefficient at exploring the space. If the focus is transfer, one could argue that another way of training the PATH net could be by training jointly the PATH net and goal net, with the intend of then transferring to another reward scheme.

A3C is known to be quite high variance. I think there are a lot of little details that don’t seem that explicit to me. How many seeds are run for each curve (are the results an average over multiple seeds). What hyper-parameters are used. What is the variance between the seeds. I feel that while the proposed solution is very intuitive, and probably works as described, the paper does not do a great job at properly comparing with baseline and make sure the results are solid. In particular looking at Riverraid-new is the advantage you have there significant? How does the game do on the original task?

The plots could also use a bit of help. Lines should be thicker. Even when zooming, distinguishing between colors is not easy. Because there are more than two lines in some plots, it can also hurt people that can’t distinguish colors easily.

---

### Decision · Program_Chairs · 2018-01-29
**ICLR 2018 Conference Acceptance Decision**

**Decision:**

Accept (Poster)

**Comment:**

All reviewers recommend accepting this paper, and this AC agrees.